# Youth Psychiatric Hospitalization in Israel during COVID-19: A Multi-Center Study

**DOI:** 10.3390/ijerph19169870

**Published:** 2022-08-10

**Authors:** Chen Dror, Nimrod Hertz-Palmor, Yael Barzilai, Schoen Gila, Bretler-Zager Tali, Gizunterman Alex, Lahav Tal, Kritchmann-Lupo Maya, Saker Talia, Gothelf Doron, Yuval Bloch

**Affiliations:** 1The Emotion-Cognition Research Center, Shalvata Mental Health Center, Hod Hasharon 45100, Israel; 2Sackler Faculty of Medicine, Tel-Aviv University, Tel Aviv 69978, Israel; 3Sheba Medical Center, Ramat Gan 52621, Israel; 4Geha Mental Health Center, Petah Tikva 49100, Israel; 5Ziv Medical Center (Safed), Safed 13100, Israel; 6The Azrieli Faculty of Medicine, Bar-Ilan University, Poriya 15208, Israel; 7Eitanim Mental Health Center, Harav Rafael Katzalbogen, Jerusalem 9097200, Israel; 8Faculty of Medicine, Hebrew University of Jerusalem, Jerusalem 9112102, Israel; 9Nes-ziona Mental Health Center, Beer Yaakov 70350, Israel

**Keywords:** COVID-19, psychiatric hospitalization, restrictive measures, internalizing disorders

## Abstract

During the COVID-19 pandemic there have been numerous reports of increases in psychiatric morbidity and a deterioration of status among existing patients. There is little information about how this increase has affected youth and rates of adolescent psychiatric hospitalization. Our study was aimed at examining trends in youth psychiatric hospitalization during the first year of the COVID-19 pandemic. **Method:** We used medical records to compare trends in hospitalization rates from 2019 to 2020, among psychiatric youth wards from five different centers in Israel. **Results:** The number of patients that were hospitalized in youth psychiatric wards decreased significantly from 2019 (*Mean ± SD*=52.2 *±* 28.6 per month) to 2020 (*M* *± SD* = 40.8 *±* 22.0; *unstandardized B* = −11.4, *95% CI* = −14.4 to −8.3, *p* < 0.0001). There was a significant decrease in the number of patients that were hospitalized due to internalizing disorders from 2019 (*M* *± SD* = 22.3 *±* 9.3 per month) to 2020 (*M* *± SD* = 16.8 *±* 7.7; *B* = −5.5, *95% CI* = −8.0 to −3.0, *p* = 0.0002) and a marginally significant increase in the number of restraints per month (2019: *M* *± SD* = 2.8 *±* 6.8, 2020: *M* *± SD* = 9.0 *±* 14.5; *Z =* −1.96, *Rosenthal’s r* = 0.36, *p* = 0.07). **Conclusions:** There was a significant decline in psychiatric hospitalizations during the pandemic, specifically among patients suffering from internalizing disorders. The reasons for this decline, and the future impact these changes had on hospitalizations during the pandemic demand further research. **Study limitations:** This is a retrospective multicenter study from five medical centers in Israel, therefore generalizability of our findings is limited.

## 1. Introduction

The COVID-19 pandemic is a major stressor which exposes entire populations to harsh lockdowns, social distancing, and fear of infection and its consequences, including a prolonged negative effect on mental health [1,2]. The ongoing situation has led to high rates of morbidity and mortality, which has in turn triggered overwhelming anxiety, fear and grief [2]. Past events such as the SARS pandemic in 2003 verified that people in quarantine and high-risk workers (e.g., physicians) are at a greater risk for mental illness [1]. To contain the spread of infection, the Israeli government ordered three lockdowns during the first year of the pandemic (March to December of 2020), each lasting five to eight weeks. During these periods, schools were closed for in-person teaching and instead operated via remote-learning for substantial periods. During these months, there were numerous reports of increases in psychiatric morbidity and a deterioration of current status among existing patients [3]. Adolescents with mental health problems may be less likely to tolerate a lockdown [4]. In a survey which included 2111 adolescents from the UK with a history of mental illness, 83% agreed that the pandemic had worsened their mental health, and 26% said that they were no longer able to access mental health support [5]. There is a global concern about the ability of health-care systems to adequately support patients with psychiatric disorders [6]. Additionally, there was a decline in children’s and adolescents’ visits to psychiatric ERs during the early stages of the pandemic [7,8]. Conversely, in a recent study we showed that the first year of the COVID-19 pandemic was no different from the continuous decade long rise of referrals to the children and adolescents’ psychiatric ER and the second year showed an additional incline beyond the general trend [9]. There has been also a rise in visits of patients with severe mental disorders such as psychotic spectrum disorders and bipolar disorder [10]. A recent study examining rates of adult psychiatric hospitalization during the pandemic in a tertiary care hospital in Northern India showed a significant decline in admissions compared to those in pre-COVID-19, but with longer durations of stay. These patients were mainly diagnosed with psychotic spectrum disorders and mood disorders, predominantly mania [11]. A previous study showed that the length of stay in psychiatric inpatient youth clinics is associated with psychiatric severity, as reflected by prescription of mood stabilizers and self-reports of internalizing symptoms [12]. The use of restrictive measures, such as restraints and seclusion, are prevalent in youth psychiatric wards worldwide when managing patients at risk of harming themselves and/or others [13]. Before the outbreak of COVID-19, there was a non-homogenous decline in the use of restrictive measures in psychiatric wards [14,15]. Changes in ward policy and practicing techniques, such as counseling/de-escalation strategies by skilled nursing staff, have made an important contribution to this decline [16,17]. Unfortunately, during the pandemic, visits and vacations were restricted due to the fear of spreading the virus. Moreover, the various measures taken to reduce the spread of COVID-19 within inpatient wards created a major burden for the staff [18]. In addition, healthcare workers were reported to suffer anxiety, depression and acute stress symptoms at high rates while coping with the pandemic [19,20,21,22]. Psychiatric hospitalization, especially among children and adolescents, is considered a last resort. It usually marks immediate danger due to a mental health crisis. Theoretically, one would expect these crisis conditions to worsen during the pandemic, but previous studies have not supported this hypothesis [23]. Our current study was aimed at examining whether the decline in hospitalization that was reported in adult psychiatric wards exists in child and adolescent wards as well. Another aim was to evaluate how the pandemic affected the hospitalization of children and adolescents with different diagnoses, and how it affected the use of restrictive measures in wards. To that end, we compared youth psychiatric hospitalization during the first year of the COVID-19 pandemic (2020), with the preceding year (2019). We used data from five youth psychiatric wards in three different centers in Israel, from March to December of each year, thereby covering the relevant timeframe of the pandemic. In line with previous reports regarding psychiatric ER’s visits, we hypothesized that there will be a decline in admissions to youth psychiatric wards during 2020. In accordance with recent studies, we expected a stable or comparatively higher admission rate of patients diagnosed with severe mental disorders (such as psychotic spectrum disorders) and longer duration of stay for admitted patients. We also predicted, due to the complexities brought on by the pandemic situation, that there would be an increased usage of restrictive measures in the daily care of patients in these wards.

## 2. Method

The study was approved by the institutional review boards (IRBs) of the five cooperating centers that provided the basis of this study: Shalvata [0015-20], Geha [07-20], Nes-Ziona [653-20], Ziv and Eitanim [13-20] mental health centers, and was promoted by the Israel Association of Child and Adolescent Psychiatry. The five centers are accountable for 155 of the total 230 [24] active psychiatric inpatient beds for children and adolescents in Israel. Three additional centers were not able to join in the current study.

### 2.1. Sample

It is important to note that the Israeli population is heterogeneous in ethno-cultural origin, religion, and type of residency, and thus probably allows for better generalization to different populations. During the pandemic there were more cases of infection among the Muslim minority and the Ultraorthodox Jewish community (partially due to less awareness of the dangers of the pandemic and more suspiciousness towards regulations set by the government) [25].

Five youth psychiatric wards in five different centers were included in the study, four are located in psychiatric centers, and one (Ziv) is part of a general hospital. All wards are considered “acute wards” and do not provide “chronic adolescent hospitalizations”. The wards’ policy is short-term protection, intensive assessment, and creating an integrative treatment plan. For long-term inpatient care, patients are referred to post-hospitalization boarding schools. The Shalvata Mental Health Center serves a population of approximately 500,000 inhabitants in the center of Israel and covers a significant portion of the Muslim population in Israel. The Geha Health Center serves a population of approximately 800,000 inhabitants and covers a comparatively large population of Ultraorthodox Jews. Nes Ziona health center has a youth ward and a children ward which covers a population of more than 1 million children. Ziv medical center serves an ethnically diverse population of the Galil and Golan heights regions. Eitanim psychiatric hospital serves the youth population of the Jerusalem area.

The current study focused on minors, aged 6–18, who were admitted to a youth psychiatric ward in one of the five above-mentioned centers between March and December 2020 (the first patient to be diagnosed with COVID-19 in Israel was on 21 February 2020) in need of acute psychiatric care. No inclusion criteria were established other than being admitted during the relevant timeframe. A comparison was made with hospital admissions during the same months in 2019.

### 2.2. Measures

#### 2.2.1. Procedure

Data were collected from the electronic files of all five centers. Measurements were extracted monthly from January 2019 to December 2020. As mentioned, given that the first case of COVID-19 in Israel was diagnosed on 21 February 2020, data from January and February of both years (i.e., 2019 and 2020) were removed from the analyses. Due to our relatively small sample, we could not analyze our data based on all ICD diagnostic categorize. Thus, like previous studies in the field, we grouped diagnoses into three clinically meaningful groups [7,8,10,11]: (a) internalizing disorders, which include depressive disorders, anxiety disorders, obsessive-compulsive and related disorders, trauma and stressor-related disorders, dissociative disorders, eating disorders, dysthymia, and somatic disorders [26]; (b) externalizing disorders, which include attention-deficit/hyperactivity disorder (ADHD), oppositional defiant disorder (ODD) and conduct disorder (CD) [27]; and (c) severe mental illness, which includes psychotic spectrum disorder, bipolar spectrum disorders, autism spectrum disorders and certain personality disorders [28]. The full list of diagnoses classified into each category is presented in the appendix (Appendix A). In child and adolescent psychiatric wards in Israel, restrictive measures are used when needed. Special protection orders are defined as orders for a close observation of the patient by a member of staff for a period of eight hours. This means that a member of the staff is with the patients in close proximity at all times during those eight hours. Each special protection order can be extended if needed based on a psychiatrist’s clinical decision. A restraint order is given in cases where patients are physically endangering themselves or others. Restraints are performed by a member of the nursing staff. They restrain the patient to a bed with assigned straps for a period of up to an hour. The restraint order can be extended for another hour by a psychiatrist and extended further only by the head of the mental health center. Restrained patients are closely monitored, and restraint orders are constantly regulated.

During this time period, if a patient or a staff member fell ill, or was “positive for COVID-19” the ward was secluded according to regulations (14 days and then another seven days), and new patients needing hospitalization were transferred to another ward.

#### 2.2.2. Data Analysis

Mixed-effects linear regression models were applied to assess differences between 2020 and 2019, with respect to comparable months (March to December of each year) and the origin of the data (i.e., the specific medical center from which data were retrieved), which were modeled as random factors. We conducted a series of univariate linear regressions, each time with the factor of interest as the dependent variable and adjusted for multiple comparisons with Bejamini and Hochberg’s false discovery rate correction [29]. In cases where observations significantly deviated from a normal distribution (tested with Kolmogorov Smirnov test and an α of <0.01), the non-parametric Wilcoxon test was applied instead. Effect sizes were estimated with unstandardized *B* coefficients and Rosenthal’s *r* [30], for variables that were tested with mixed-effects regression and Wilcoxon test, respectively. We assessed which sociodemographic and clinical properties might be associated with hospitalization rates using a similar methodology of mixed-effects models, where in the first model age and gender were modeled as fixed effects, and in the second we modeled diagnoses (internalizing, externalizing and other severe diagnoses) as fixed effects. In both models, the origin of the data (i.e., medical center) and the dates of hospitalization were modeled as random effects. We employed two separate models instead of one since we had missing data for some of the centers. Thus, the sociodemographic model included Geha, Nes Ziona and Eitanim, while the clinical model included Geha and Shalvata.

We used the standard chance of a type-I error set at <0.05 and adjusted for multiple comparisons as described. The analysis was conducted with the ‘lmerTest’ and ‘stats’ packages in R [31].

## 3. Results

There was a significant reduction in the number of patients that were admitted to psychiatric wards from 2019 (*Mean ± SD* = 52.2 *±* 28.6 per month) to 2020 (*M ± SD* = 40.8 *±* 22.0; *unstandardized B* = −11.4, *95% CI* = −14.4 to −8.3, *p* < 0.0001). In line with this, there were significant decreases in the number of full hospitalizations (2019: *M ± SD* = 47.5 *±* 29.5 per month, 2020: *M ± SD* = 38.5 *±* 22.9; *B =* −8.9, *95% CI* = −11.8 to −6.1, *p* < 0.0001), day hospitalizations (2019: *M ± SD* = 5.3 *± 4.0*, 2020: *M ± SD* = 2.6 *±* 3.3; *Z =* −4.04, *Rosenthal’s r* = 0.57, *p* = 0.0001) and the cumulative sum of hospitalization days per month (2019: *M ± SD* = 763.9 *±* 568.5, 2020: *M ± SD* = 651.5 *±* 489.4; *B =* −112.4, *95% CI* = −163.3 to −61.5, *p* = 0.0001). The mean duration of the hospitalization period remained unchanged between 2019 and 2020. There was a significant decrease in the number of patients that were admitted due to internalizing disorders from 2019 (*M ± SD* = 22.3 *±* 9.3 per month) to 2020 (*M ± SD* = 16.8 *±* 7.7; *B* = −5.5, *95% CI* = −8.0 to −3.0, *p* = 0.0002), and only a marginally significant decrease in the number of patients that were admitted due to externalizing disorders (2019: *M ± SD* = 22.6 *±* 11.2, 2020: *M ± SD* = 19.3 *±* 8.5; *B =* −3.3, *95% CI* = −6.7 to 0.1, *p* = 0.07) and severe psychiatric disorders (2019: *M ± SD* = 48.9 *±* 17.7, 2020: *M ± SD* = 42.6 *±* 12.6; *B =* −6.3, *95% CI* = −12.1 to −0.5, *p* = 0.06).

There was a significant decrease in the number of special protection orders (2019: *M ± SD* = 742.9 *±* 164.2 per month, 2020: *M ± SD* = 44.8.8 *±* 150.5; *B =* −294.1, *95% CI* = −431.6 to −156.6, *p* = 0.001) and a significant increase in the mean duration of special protections (2019: *M ± SD* = 4.6 *±* 1.2 h 2020: *M ± SD* = 7.8 *±* 0.6; *B =* 3.1, *95% CI* = 2.3 to 4.0, *p* < 0.0001). There was no change in the number of patients toward which special protection orders were directed, and only a marginally significant increase in the number of restraints per month (2019: *M ± SD* = 2.8 *±* 6.8, 2020: *M ± SD* = 9.0 *±* 14.5; *Z =* −1.96, *Rosenthal’s r* = 0.36, *p* = 0.07). The results are summarized in Table 1. Age was associated with more hospitalizations (*B* = 9.38, *95% CI* = 2.73 to 15.21, *p* = 0.009), meaning that an increase of one year in the child’s age group predicted ~9 more hospitalizations per month in comparison with children who are one year younger. Gender was not associated with the number of hospitalizations. Examining clinical properties, we observed that hospitalization of patients with severe mental disorders did not change between 2019 and 2020. On the other hand, in 2020 there was an increase in the proportion of patients hospitalized due to internalizing disorders (*standardized β* = 0.37, *95% CI* = 0.17 to 0.57, *p* = 0.003) and externalizing disorders (*β* = 0.19, *95% CI* = 0.03 to 0.35, *p* = 0.038), out of the total number of hospitalizations. However, as showed above, there was a decrease in the absolute number of hospitalizations of patients diagnosed with those disorders.

## 4. Discussion

Our study examined trends in youth psychiatric hospitalizations during the first year of the COVID-19 pandemic. We explored admissions to youth psychiatric wards from five different centers in Israel from March to December 2020, and compared it to the corresponding period in 2019. We observed a significant reduction in hospitalizations in youth psychiatric wards from 2019 to 2020. This reduction in hospitalizations is in line with a recent study that showed a reduction in child and adolescents’ ER admissions and hospitalizations during the first months of the COVID-19 pandemic at two urban university hospitals in Turin and Rome, Italy [7]. Another study reported a 61% decrease in pediatric mental health visits to the ER at the Yale New Haven Children’s Hospital [8]. During the first months of the pandemic, the decision to hospitalize patients was based not only on psychiatric severity but also on the intent to avoid exposure of patients to the risk of COVID-19 infection in the wards, and to allocate hospital resources towards the most complicated patients [8]. Moreover, in order to minimize the risk for exposure of hospitalized patients to COVID-19 infection, visitations of patients’ family members were either restricted or prohibited in the wards. These restrictions may have led patients and their parents to be more reluctant and even to refuse hospitalization compared to pre-pandemic periods. It might have also led to greater distress and frustration for hospitalized patients who did not meet their family for a long period of time and as a result may have increased aggressive behaviors, which can explain the increase in the number of restraints shown in our study. We also observed a significant decrease in the number of patients that were hospitalized due to internalizing disorders from 2019 to 2020. This result may be related to the fact that during lockdowns there was a suspension of most academic, social and recreational activities, which may have alleviated mental stress in children and adolescents who suffer from separation anxiety, social anxiety, and school refusal [32]. In addition, patients suffering bullying, academic difficulties and other stressors may have experienced home-schooling as relieving [23,32]. On the other hand, some patients experienced a great burden because of the lockdowns and the reduced opportunities for social recreational activities and outdoor physical exercise, which may have worsened their mental health [4,7]. Also, some patients diagnosed with internalizing disorders may have suffered greater anxiety due to the fear of infection and uncertainty about the future during the early months of the pandemic [3]. It is possible that although there was a greater need to seek mental health care, patients and their families were hesitant because of the fear of infection and lack of alternatives to full hospitalization. Another potential reason for the reduction in hospitalizations, especially in regard to internalizing disorders, is greater parental involvement and supervision due to more remote working from home, which may have contributed to greater family cohesion [7]. This proposed greater parental supervision at home may have helped avoiding hospitalization for protection reasons. However, we did not observe a major decline in hospitalization due to externalizing disorders or severe psychiatric disorders. We assume that among patients who tend to get more violent and require close supervision and more intensive psychiatric care, the decision to hospitalize them was made despite the COVID-19 regulations in the wards and the risk of infection during the pandemic period. This difference in hospitalization rates between patients coping with internalizing disorders and those coping with externalizing and severe mental illness during the pandemic condition may indicate which patients need care at an inpatient facility and which might be hospitalized by default because there are no other reasonable alternatives. We also showed that an older age predicted a higher chance for hospitalization as shown by previous studies [33,34]. This raises questions about what mental health care younger children are offered as an alternative to hospitalization. Recent studies addressing the mental health crisis developing in the current pandemic acknowledge the need for additional and sustained investment in virtual as well as face-to-face psychological and other mental health services, especially for children and adolescents [35]. Triaging of resources becomes necessary as health care capacity is outstripped by demand and developing efficacious and available alternatives to psychiatric hospitalization is crucial [36]. Contrary to our hypothesis, we observed that the use of restraints did not significantly increase during 2020 in our youth wards. This may be related to the growing use of counseling/de-escalation strategies by skilled nursing staff members, despite their pandemic induced exhaustion [16,19,20]. The decrease in the use of special protection in the wards may be explained by the decrease in the hospitalization rate allowing the ward staff to provide more personal care and attention for each patient, when compared to periods of full capacity.

## 5. Limitations

Our study had a few limitations. Clinical data were collected from the medical files of the patients and were not assessed systematically using standardized rating scales or interviews. Our study relates to the changes in practice during the COVID-19 pandemic, but we could not assess psychiatric patients who did not seek mental-care or exploited alternative options in the community. We did not have access to information about involuntary hospitalization rates in the wards and thus could not evaluate its correlation with restriction measurement use, diagnosis of patients and other hospitalization related factors. Our sample is relatively small, which limited our ability to categorize patients based on diagnosis due to small group numbers for certain categories. However, the fact that the patients were consistently diagnosed and treated in the wards strengthens the validity of our findings. Another limitation is the fact that our study is a retrospective study and as such limits our ability to retrieve more data about the patients and their parents, which could be relevant to better understand the reasons for the results shown in this study.

## 6. Conclusions

Our study showed a significant reduction in admissions to youth psychiatric wards from 2019 to 2020, specifically among patients with internalizing disorders. Future studies should examine alternative options for patients who do not necessarily need hospitalization or refuse it and could benefit from home hybrid outpatient care programs that include both frontal and remote therapeutic care.

## Figures and Tables

**Table 1 ijerph-19-09870-t001:** Comparison of main characteristics in psychiatric hospitalization of youth between 2019 and 2020.

Variable	Observations	Mean (SD)		Β (95% CI)		*p* ^a^
		2019	2020			
Number of patients	100	52.2 (28.6)	40.8 (22.0)	−11.4 (−14.4, −8.3)		<0.0001
Number of patients—full hospitalization only	100	47.5 (29.5)	38.5 (22.9)	−8.9 (−11.8, −6.1)		<0.0001
Cumulative sum of hospitalization days	100	763.9 (568.5)	651.5 (489.4)	−112.4 (−163.3, −61.5)		0.0001
Patients with internalizing disorders	40	22.3 (9.5)	16.8 (7.7)	−5.5 (−8.0, −3.0)		0.0002
Patients with externalizing disorders	40	22.6 (11.2)	19.3 (8.5)	−3.3 (−6.7, 0.1)		0.07
Patients with other severe disorders	40	48.9 (17.7)	42.6 (12.6)	−6.3 (−12.1, −0.5)		0.06
Special protections—number of patients	20	21.0 (4.4)	21.2 (2.9)	0.2 (−3.0, 3.4)		0.91
Special protections—number of orders	20	742.9 (164.2)	448.8 (150.5)	−294.1 (−431.6, −156.6)		0.001
Special protections-mean duration	20	4.6 (1.2)	7.8 (0.6)	3.1 (2.3, 4.0)		<0.0001
**Non-parametric testing ^b^**				**Statistic**	**Effect size**	
Number of patients—day hospitalization only	100	5.3 (4.0)	2.6 (3.3)	*Z* = −4.04	*r* = 0.57	0.0001
Mean hospitalization duration	80	35.9 (25.0)	35.1 (19.3)	*Z* = −0.29	*r* = 0.05	0.84
Number of restraints	60	2.8 (6.8)	9.0 (14.5)	*Z* = −1.96	*r* = 0.36	0.07

Across all analyses, nested factors included the medical center and month of observation. ^a^ Adjusted after False Discovery Rate correction. ^b^ Wilcoxon signed rank test, applied for variables whose distribution violated the normality assumption at a chance of α < 0.01 according to Kolmogorov-Smirnov test.

## Data Availability

Data will be shared upon request.

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
