# Peer review of "Youth Psychiatric Hospitalization in Israel during COVID-19: A Multi-Center Study"

_ijerph, 2022, doi:10.3390/ijerph19169870_

Round 1

Reviewer 1 Report

Many thanks to the editor and authors for the opportunity to review this manuscript. The paper is well-written and is of interest for the journal. However, I would propose some minor changes prior to its publication:

- First of all, I would appreciate it if the results were presented in tables (I haven´t been able to see them in the document).

- in the methodology, hospitalizations are disthinguished into three categories: externalizing disorder, internalizing disorder, and severe mental illnes. I would appreciate if the autors argue this distinction. 

-  Finally, I think it would be interesting , in the discussion section, for the autors to explain what they think about the different impact of the pandemic on the trend towards hospitalization according to the diagnostic category. 

Author Response

Reviewer 1

Remark 1: First of all, I would appreciate it if the results were presented in tables (I haven´t been able to see them in the document).

Answer 1: We thank the reviewer for his comment. A table depicating the main results was attaced to the original manuscript and we now intgated in the revised manuscript itself.

Remark 2:  In the methodology, hospitalizations are disthinguished into three categories: externalizing disorder, internalizing disorder, and severe mental illnes. I would appreciate if the autors argue this distinction. 

Answer 2: We thank the reviewer for his comment and agree that we should better explain the use of those diagnostic categories in the study. See revised manuscript page 3, lines 130-132: "Due to our relatively small sample and since previous studies showed different trends of psychiatric hospitalization in patients diagnosed with one of the following three categories we decided to group our participants into those categories (7-10)."

Remark 3: Finally, I think it would be interesting , in the discussion section, for the autors to explain what they think about the different impact of the pandemic on the trend towards hospitalization according to the diagnostic category. 

Answer 3: We agree with the reviewer that it is of intrest to discuss the different unfluance of thr COVID-19 pandemic on trends towards hospitalization according to the daignostic categories and we did extensively discussed it in the manuscript. See page 8, lines 238-264:  "We also observed a significant decrease in the number of patients that were hospitalized due to internalizing disorders from 2019 to 2020. This result may be related to the fact that during lockdowns there was a suspension of most academic, social and recreational activities, which may have alleviated mental stress in children and adolescents who suffer from separation anxiety, social anxiety, and school refusal (31). In addition, patients suffering bullying, academic difficulties and other stressors may have experienced home-schooling as relieving (22,31). On the other hand, some patients experienced a great burden because of the lockdowns and the reduced opportunities for social recreational activities and outdoor physical exercise, which may have worsened their mental health (4,7). Also, some patients diagnosed with internalizing disorders may have suffered greater anxiety due to the fear of infection and uncertainty about the future during the early months of the pandemic (3). It is possible that although there was a greater need to seek mental health care, patients and their families were hesitant because of the fear of infection and lack of alternatives to full hospitalization. Another potential reason for the reduction in hospitalizations, especially in regards to internalizing disorders, is greater parental involvement and supervision due to more remote working from home, which may have contributed to greater family cohesion (7). This proposed greater parental supervision at home may have helped avoiding hospitalization for protection reasons. However, we did not observe a major decline in hospitalization due to externalizing disorders or severe psychiatric disorders. We assume that among patients who tend to get more violent and require close supervision and more intensive psychiatric care, the decision to hospitalize them was made despite the COVID-19 regulations in the wards and the risk of infection during the pandemic period. This difference in hospitalization rates between patients coping with internalizing disorders and those coping with externalizing and severe mental illness during the pandemic condition may indicate which patients need care at an inpatient facility and which might be hospitalized by default because there are no other reasonable alternatives."

Reviewer 2 Report

The brief report by Dror et al. presents an interesting retrospective study, reflecting on the effects of COVID pandemic on mental and public health.

Study design is reliable, statistical analysis is correct. However, their sample size is not huge, but they collected data from medical centers accountable for 67.39 % of all active psychiatric inpatient beds for children and adolescents in Israel. A further advantage is that these centers are located in heterogenous demographic areas. Only disadvantage, that they didn't have data about involuntary admissions, which lowers the validity of results regarding severe mental illnesses. However, given all their limitations, they collected an appropriate dataset. The authors categorized patients into three diagnostic groups: internalizing, externalizing disorders, and severe mental illness. In my opinion, these are valid categories. I have to mention, that I didn't have access to the cited Appendix A, so I could not review the full list of diagnoses. The results are clearly presented, although I could not see Table 1, it was not in the manuscript. The last sentence of the results section (rows 203-207) seemed unclear to me, and I suggest to modify or at least explain what does it mean, because in its current form, it can confuse the reader. The Discussion is well-written and authors drawn good consequences from their results. Limitations are also accurately declared. 

In summary, I suggest only a minor revision in one part of the results as I mentioned above. 

Author Response

Reviewer 2

Remark 1: The brief report by Dror et al. presents an interesting retrospective study, reflecting on the effects of COVID pandemic on mental and public health. Study design is reliable, statistical analysis is correct. However, their sample size is not huge, but they collected data from medical centers accountable for 67.39 % of all active psychiatric inpatient beds for children and adolescents in Israel. A further advantage is that these centers are located in heterogenous demographic areas. Only disadvantage, that they didn't have data about involuntary admissions, which lowers the validity of results regarding severe mental illnesses. However, given all their limitations, they collected an appropriate dataset. The authors categorized patients into three diagnostic groups: internalizing, externalizing disorders, and severe mental illness. In my opinion, these are valid categories. I have to mention, that I didn't have access to the cited Appendix A, so I could not review the full list of diagnoses. The results are clearly presented, although I could not see Table 1, it was not in the manuscript.

Answer 1: We thank the reviewer for his comment. A table depicating the main results was attaced to the original manuscript as well as appedices describing the full list of dignosis of participants and we now intgated in the revised manuscript itself.

Remark 2: The last sentence of the results section (rows 203-207) seemed unclear to me, and I suggest to modify or at least explain what does it mean, because in its current form, it can confuse the reader.

Answer 2: We agree with the reviewer that this mentioned sentence can be confusing and we modified to be and calrified its meaning in the revised manuscript in page 5, lines 206-2011: " On the other hand, in 2020 there was an increase in the proportion of patients hospitalized due to internalizing disorders (standardized β=0.37, 95% CI=0.17 to 0.57, p=.003) and externalizing disorders (β=0.19, 95% CI=0.03 to 0.35, p=.038), out of the total number of hospitalizations. However, as showed above, there was a decrease in the absolute number of hospitalizations of patients diagnosed with those disorders".